# Comparison of Ergogenic Effects of Caffeine and Nitrate Supplementation on Speed, Power and Repeated Sprint Performance of Soccer Players

**Dimitris Karampelas** [1], **Konstantinos Antonopoulos** [2], **Yiannis Michailidis** [1,*], **Michalis Mitrotasios** [3], **Athanasios Mandroukas** [4] **and Thomas Metaxas** [1]

1   Laboratory of Evaluation of Human Biological Performance, Department of Physical Education and Sport Sciences, Aristotle University of Thessaloniki, 57001 Thessaloniki, Greece; dikarampe@phed.auth.gr (D.K.); tommet@phed.auth.gr (T.M.)
2   Chemical Engineering Department, Aristotle University of Thessaloniki, 57001 Thessaloniki, Greece; konstanda@cheng.auth.gr
3   Department of Sport Science, National and Kapodistrian University of Athens, 35611 Athens, Greece; micmit@phed.uoa.gr
4   Physical Education and Sport, Charles University, 12800 Prague, Czech Republic; amandrou@phed.auth.gr
*   Correspondence: ioannimd@phed.auth.gr; Tel.: +30-23-1099-2233

**Abstract:** Caffeine and nitrates have both been reported to enhance performance in power efforts; however, it is not clear which supplement is most effective. The aim of this study was to compare the effects of caffeine and nitrates on the performance of semi-professional soccer players during different fitness tests. Ten male soccer players in a randomized crossover design were assigned to receive caffeine (5 mg/kg body mass) (CG), nitrate ((250 mL/150 mg of $NO_3^-$) (NG), or a placebo (PG) on three different occasions. In each treatment, the participants performed the following tests: 10 m and 30 m sprints, the Illinois agility test, a countermovement jump test, a squat jump test, and a repeated sprint test (6 × 40 m). Caffeine boosted performance in jumps (CMJ: CGvsPG, $p = 0.018$; SJ: CGvsPG, $p = 0.045$ and CGvsNG, $p = 0.001$) and limited the decrease in performance in the RSA test (CGvsPG, $p = 0.012$). Nitrates limited the decrease in performance in the RSA test (NGvsPG, $p = 0.035$). In conclusion, the two supplements limited the decrease in performance in the test of repeated sprints, with caffeine showing a greater effect. Among the other tests, only caffeine improved performance, and only in the jumps. Thus, we can conclude that supplementation with caffeine 1 h before these kinds of activities at a dosage of 5 mg/kg of body weight can enhance performance.

**Keywords:** caffeine; nitrate; soccer; performance; power

## 1. Introduction

Soccer is an intermittent sport that includes low- and high-intensity actions. In a high-level soccer match, players cover 9 to 12 km [1] by walking or running at different speeds. The distance covered by high-intensity running is crucial to the performance of players and constitutes 8% to 12% of the total distance [1]. Previous studies have indicated that soccer players perform between 17 and 81 sprints in each match [2]. Their average duration is 2 to 4 s, and most of them (>90%) are less than 20 m [2]. It is obvious that high-intensity actions are particularly important for performance in soccer.

The use of dietary supplements is particularly common in the field of sport to enhance performance in maximum efforts [3]. However, of the existing supplements, only some appear to have an ergogenic effect confirmed by scientific studies [3]. Two of these substances are caffeine and nitrates from beetroot juice. The effect of caffeine on athletic performance has been studied for the last 30 years, whereas the effects of nitrates have been investigated for the past 10 years.

As mentioned above, caffeine is one of the most popular ergonomic aids in both endurance and power sports [4]. Its possible mechanisms of action are considered to enhance the oxidation of fats, resulting in the saving of carbohydrates [5], reduction in the rating of perceived exertion (RPE) due to adenosine antagonism in the central nervous system [6], and the activation of more kinetic units [4]. However, its effect is controversial; in addition to studies that found performance enhancement [7], there are also studies in which no significant effects were observed [8].

Over the past decade, several studies have focused on the effect of nitrates on athletic performance [9]. Nitrates in the human body are converted into nitric oxide (NO), which has hemodynamic and metabolic effects on the body. It can cause vasodilation by lowering blood pressure and increasing the transfer of gases and nutrients to the exercised muscles [10]. From the literature, it appears that the above physiological effects can enhance performance [9]. However, there are also studies where performance was unaffected after taking nitrates [11].

As mentioned above, performance in soccer depends on actions taken at high intensity. Therefore, it is especially important to maintain the capacity for high-intensity action throughout a soccer match. Both caffeine and nitrates have been reported to improve performance during anaerobic effort. The methodological differences of different studies can influence their results, making it difficult to compare studies. There are also no studies comparing the effects of caffeine and nitrates on the performance of soccer players during power tests. The purpose of this study is to compare the effects of caffeine and nitrates on semi-professional soccer players. We hypothesized that caffeine and nitrate ingestion would provoke dissimilar performance effects.

## 2. Material and Methods

### 2.1. Design

Participants performed activities in three conditions that were at least five days apart from each other. In a randomized crossover design, subjects were then assigned to receive a placebo (PG), caffeine (CG) (5 mg/kg), or nitrate (NG) (250 mL/150 mg of $NO_3^-$). Both the caffeine dose and the nitrate dose used have been reported to have ergogenic effects on athletic performance.

Each day of measurement was preceded by two days of wash-out. During the participants' first visit, anthropometric and height, weight, and body fat measurements were performed. The design of the study is presented in Figure 1. After a standard warm-up lasting 15 min, a squat jump (SJ) and countermovement jump test (CMJ) were carried out, followed by 10 m and 30 m speed tests and the Illinois agility test. Finally, a repeated sprint test (RSA) [12] was carried out. The measurements were performed on an open soccer field with synthetic turf. The study was carried out during the season by modifying the contents of the participants' training sessions.

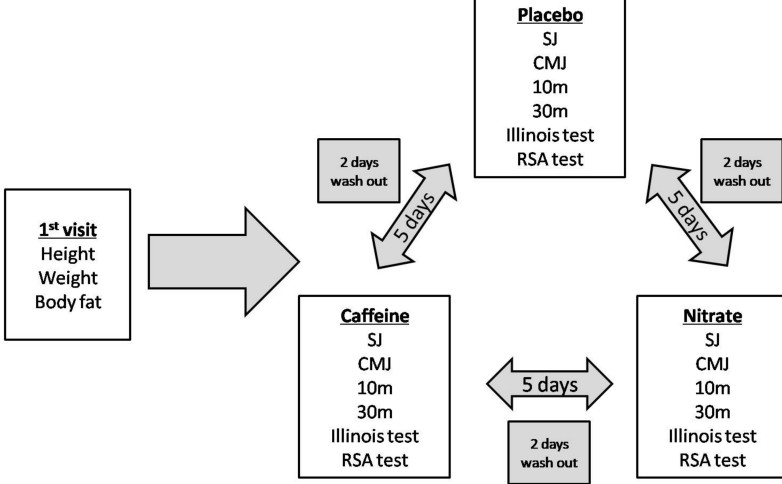

**Figure 1.** Study design.

## 2.2. Subjects

Ten healthy semi-professional male soccer players took part in the study voluntarily. The study was approved by the Ethics Committee of the School of Physical Education and Sport Science at Thessaloniki (66/2021) and conformed to the Helsinki Declaration. All of the participants were members of soccer teams, trained five days a week and participated in one match a week. The participants were informed about the details of the study and signed the corresponding written consent form. In addition, they completed questionnaires about their medical history and caffeine consumption. The characteristics of the participants are presented in Table 1.

**Table 1.** Anthropometric characteristics of participants.

|  | Mean $\pm$ SD [a] |
| --- | --- |
| Age (years) | $21.3 \pm 0.9$ |
| Height (cm) | $176.1 \pm 3.8$ |
| Weight (kg) | $73.49 \pm 4.44$ |
| Body fat % | $12.68 \pm 3.09$ |
| Body mass index (kg/cm$^2$) | $23.65 \pm 1.54$ |
| Training age (years) | $14.08 \pm 2.02$ |

[a] SD: Standard deviation.

## 2.3. Anthropometric Measurements

An electronic digital weight scale and a height scale (Seca 220e, Seca, Hamburg, Germany) were used to measure the body mass and height of the players. These two measurements had an accuracy of 0.1 kg and 0.1 cm in the respective evaluations. During the measurements, the participants were barefoot and wore only underwear. To assess body fat, a Lafayette skinfold caliber (Sagamore, Lafayette, Ins. Co., Indiana) was used to measure the thickness of the soccer players' hypodermic fat in four of their skinfolds (biceps, triceps, suprailiac, subscapular). All skinfold measurements were taken on the right side of the body, and the body fat percentage was calculated with the use of the Siri equation (1956) [13].

## 2.4. Speed Testing (10 m and 30 m)

A 30 m sprint test with 10 m splits (0–10 m was measured as well) was used to measure speed performance. After a 5 s countdown, the participants ran in front of three infrared photoelectric gates (Microgate, Bolzano, Italy) that recorded the times at each gate. The participants sprinted from a standing starting position with the toe of the front foot approximately 0.3 m behind the first gate. Photocells were placed 0.6 m above the ground (approximately at hip level) to capture the movement of the trunk rather than a false signal due to a limb motion. The coefficient of variation for test–retest trials was 3.1%.

## 2.5. RSA Test (Repeated Sprint Ability Test)

The RSA test consists of 6 × 40 m sprints (20 + 20 m) with a 20 s break in between sprints [12]. The athletes started from the starting line, sprinting for 20 m, stepping on a line, and returning to the starting point as quickly as possible. They followed with 20 s of recovery before starting the next sprint. In the last 5 s of recovery time, there was a countdown to the athlete being ready to start at the end of the 20 s. At the starting line, there were photoelectric gates (photocells) (Microgate, Bolzano, Italy) to record the time of each sprint. Photocells were placed 0.6 m above the ground (approximately at hip level) to capture the movement of the trunk rather than a false signal due to a limb motion. The coefficient of variation for test–retest trials was 4.1%.

## 2.6. Vertical Jump Testing

The participants performed two jump tests: (a) SJ: participants, from a stationary semi-squatted position (90° angle at the knees), performed a maximal vertical jump (VJ); (b) CMJ:

participants, from an upright standing position, performed a fast preliminary motion downwards by flexing their knees and hips followed by an explosive upward motion by extending their knees and hips. Both tests were performed with the arms akimbo. The VJ height was measured with ChronojumpBoscosystem equipment (Chonojump, Barcelona, Spain). The coefficients of variation for test–retest trials were 2.8% and 3.6% SJ and CMJ, respectively.

### 2.7. Illinois Agility Test

The Illinois agility test was set up with four markers forming a square area of 10 × 5 m. The start and finish gates were positioned at two consecutive angles of a square area, with two markers positioned on the opposite side to indicate the two turning points. Four other markers were in the center, an equal distance apart (3.1 m). Each participant had to run as quickly as possible from the start gate, follow a planned route, and slalom through the markers without knocking them down or cutting over them. From a standing position, each athlete sprinted 10 m on command and returned to the starting line, then had to swerve in and out of the markers, perform another sprint of 10 m, and complete the test by running to the finish gate. The photocells at the start and finish gates recorded the test time. The better time of two attempts was considered the Illinois agility test score. A graphic representation of the test is shown in Figure 2.

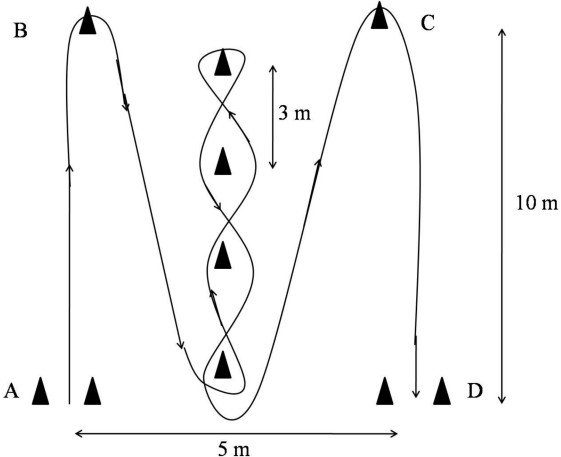

**Figure 2.** Graphical representation of Illinois agility test.

### 2.8. Statistical Analysis

Descriptive statistics (mean ± standard deviation, SD) were calculated for each parameter. Data normality was verified with the Kolmogorov–Smirnoff test. A one-way analysis of variance (ANOVA) was used to compute any differences in the subjects' performance on the tests. Wherever a significant difference was found, post hoc Bonferroni was applied. The level of significance was set at $p < 0.05$. Effect sizes (ESs) were estimated by calculating partial eta squared and were classified as small (0.01 to 0.058), medium (0.059 to 0.137) or large (0.138 or higher) according to Cohen (1988) [14]. SPSS version 25.0 was used for all analyses (SPSS Inc., Chicago, IL, USA).

### 3. Results

No differences between the three treatments were observed in the 10 m acceleration test (F = 1.343, $p = 0.293$, $\eta^2 = 0.161$) or the 30 m speed test (F = 5.512, $p = 0.185$, $\eta^2 = 0.195$); the results are presented in Figure 3A,B. In the Illinois agility test, there was no statistically significant difference between the treatments (F = 1.353, $p = 0.302$, $\eta^2 = 0.213$); the results are presented in Figure 3C.

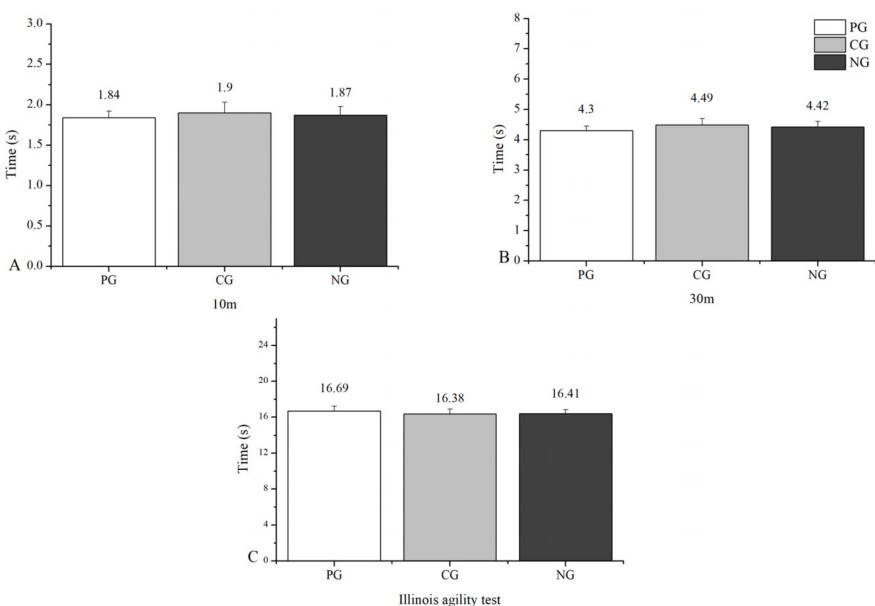

**Figure 3.** (**A**) Sprint, 10 m; (**B**) sprint, 30 m; (**C**) Illinois test.

Differences were observed in the CMJ test between the three treatments (F = 3.937, $p$ = 0.044, $\eta^2$ = 0.360). Post hoc testing showed that PG was different with CG ($p$ = 0.018). Similarly, the performance of the three treatments in the SJ was different (F = 8.325, $p$ = 0.004, $\eta^2$ = 0.543). Differences were observed between PG and CG ($p$ = 0.045) and between NG and CG ($p$ = 0.001). Results for the jump tests are presented in Figure 4.

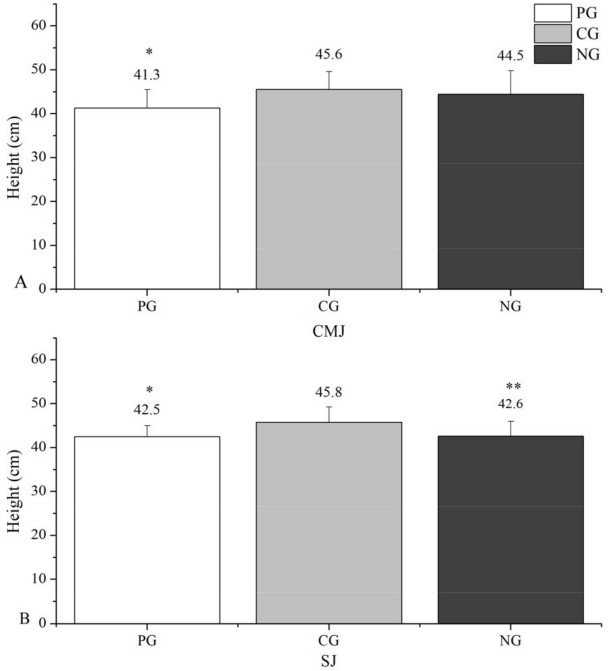

**Figure 4.** (**A**) Countermovement jump test; (**B**) squat jump test. * Denotes a significant difference between the placebo (PG) and caffeine (CG). ** Denotes a significant difference between caffeine (CG) and nitrate (NG).

No differences were observed for RSAbest and RSAmean between the treatments (best sprint: F = 0.778, $p$ = 0.485, $\eta^2$ = 0.135; mean time: F = 0.749, $p$ = 0.498, $\eta^2$ = 0.130). However, for the performance decrement during RSA test, we observed significant differences between the treatments (F = 2.153, $p$ = 0.048, $\eta^2$ = 0.301). More specifically, the differences

were observed between PG and CG (*p* = 0.012) and between PG and NG (*p* = 0.035). Results for the jump tests are presented in Figure 5.

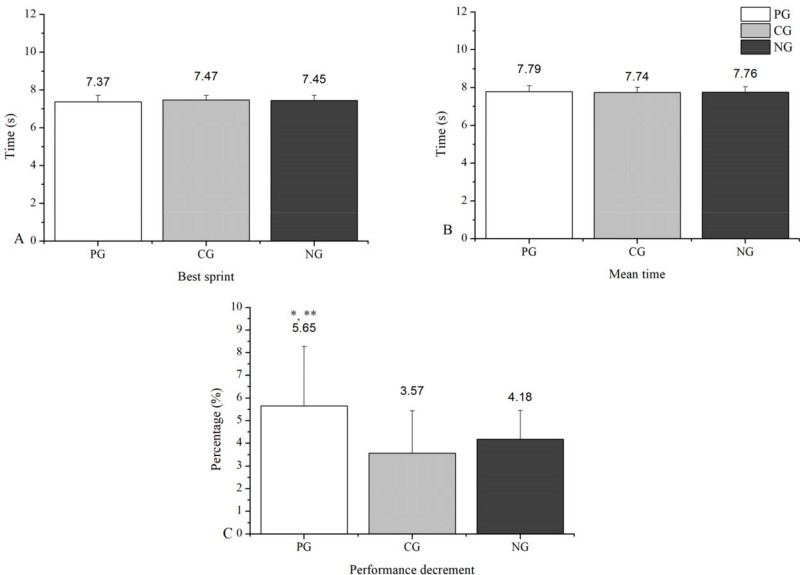

**Figure 5.** (**A**) RSA best sprint; (**B**) RSA mean time; (**C**) RSA percentage of performance decrement. * Denotes a significant difference between the placebo (PG) and caffeine (CG). ** Denotes a significant difference between the placebo (PG) and nitrate (NG).

## 4. Discussion

The results of this study showed that the caffeine supplement at a dose of 5 mg/kg of body weight improves jump performance (CMJ, SJ) and helps maintain performance in repeated sprint tests. The only ergogenic effect observed for the nitrate supplement was to maintain performance in the test of repeated sprints, with this effect being smaller, however, compared to caffeine.

No effect was observed after taking caffeine supplements in the speed or acceleration tests. These findings are in line with previous recent studies related to the effects of caffeine on soccer players [15,16]. However, there are also studies that have reported an improvement in performance in the 20 m sprint after taking caffeine [17]. The literature review shows that the number of studies which have carried out individual speed tests is minimal, because most studies use repeated sprint tests and take into account the fastest repetition to assess speed [18,19]. The effect on repeated sprints will be discussed below.

As was the case with caffeine, nitrates did not affect sprint performance. A review of the literature shows that some studies have observed an improvement in speed [20], whereas others did not notice any effect of nitrates on speed [21].

In this study, performance in the Illinois agility test was not affected by caffeine intake. This finding is consistent with an earlier study using a reaction agility test [15] and contradicts a more recent study by Ellis, Noon, Myers and Clarke (2018) [17]. More specifically, Jordan et al. (2014) [15] used a reaction agility test and found that the caffeine supplement improved reaction time on the non-dominant side of players but did not affect sprint time. In contrast, Ellis et al. (2018) [17], applying the arrowhead agility test, observed an improvement in performance after taking caffeine.

As far as nitrate supplements are concerned, no effect on the performance of soccer players in the Illinois agility test was observed. This finding is in line with the results of recent studies in tennis players and basketball players that applied the *T*-test and the 5-0-5 agility test [21,22].

In the jump tests, we found that caffeine intake improved performance in both CMJ and SJ. In SJ, the performance of the players after taking caffeine was significantly better than the other two conditions (placebo and nitrate). A review of the literature on the

effects of caffeine shows that our findings are consistent with the findings of previous research [7,13,23].

In contrast, the nitrate supplement did not cause any effect on performance in the jump tests. Our finding is consistent with the findings of previous research [24].

The repeated sprint test indicated that caffeine intake had no effect on the best sprint performance or the average time of six sprints. However, it appeared to significantly limit the decrease in performance during the test. In terms of the best sprint time, a lack of impact has been reported by other researchers [15,18]. Carr et al. (2008) [25] reported that performance improved in some of the sets of their repeated speed test. Gant et al. (2010) [26] also observed that caffeine improved performance in the Loughborough intermittent shuttle test, and Del Coso et al. (2012) [19] reported that the average speed of movement in a $7 \times 30$ m sprint test increased.

As with the caffeine supplement, nitrates did not affect the time of the best sprint or the average time of the six test sprints. However, they limited the decline in performance compared to the control condition. Recent literature shows that nitrates can increase mean and maximum power [27] and maximum exercise time [9] and improve sprint times during tests [28].

As mentioned above, the ergogenic effects of caffeine supplements have been attributed to several mechanisms [6]. Specifically, caffeine acts on the central nervous system as an adenosine agonist, inhibiting the negative effects of adenosine on neurotransmission, excitation, and pain perception [29]. The decrease in pain perception can enhance the effort during exercise that induces pain [6], and increase the firing rates of the motor units that produce forceful muscle contraction [6]. In intense efforts with longer duration, caffeine can also save glycogen by increasing the use of free fatty acids [30]. Caffeine can also enhance muscle contraction by increasing the propriety of the $Na^+/K^+$ pump and the use of $Ca^{+2}$ ions [6]. In this study, the ergogenic effect of caffeine supplements on jumps could be attributed to improving the better use of $Ca^{+2}$ [31]. With regard to the test of repeated sprints, the limitation of the decrease in performance can be attributed to the decrease in the sense of fatigue caused by caffeine and to the increase in the firing rates of the motor units [6].

Dietary nitrates are converted into nitric oxide (NO) in the body [32]. NO plays an important role in several signaling pathways of the body related to exercise, such as vasodilation, mitochondrial respiration and muscle contraction [33]. More specifically, the high availability of NO can increase muscle oxygenation and improve muscle metabolism and contraction [33,34]. It has also been shown to affect muscle fiber type II more than type I [35]. In this study, the positive effects of nitrates were shown only in the test of repeated sprints, where they helped to limit the decrease in performance. In such efforts the limited energy supply to the muscles can be a limiting performance factor. NO can limit the required energy of effort on ATP and the use of phosphocreatine [36]. This was probably the mechanism of action of the nitrates. However, no ergogenic action was observed in the speed and power tests. This can be explained by an observation from previous research which states that although nitrates can enhance the extension and contracting of skeletal muscle, these effects do not translate into any significant changes in maximum or explosive voluntary force production [34].

This study is the first to compare the ergogenic effect of these two substances during specific anaerobic tests on soccer players. However, there are also some limitations to the study, such as the small sample used, which does not allow us to generalize the results. The changes in plasma concentrations of caffeine and nitrates ($NO_2^-$ and $NO_3^-$) after taking the supplements were also not measured to confirm the effect of supplements.

## 5. Conclusions

In conclusion, acute nitrate supplementation leads to a fatigue resistance improvement during repeated sprints under controlled situations. However, it did not improve sprinting and jumping performance. Caffeine supplementation limited the decrease in performance

during the test of repeated sprints, showing a greater effect than nitrates. In addition, caffeine increased jump height which may represent a meaningful improvement for soccer players when competing for a ball in the air. Factors such as dosage and the chronic or immediate intake of supplements can affect their ergogenic effect. This requires more studies to clarify these factors but also to study the simultaneous intake of caffeine and nitrates in the performance of soccer players. In this study, a supplement of caffeine (5 mg/kg body weight) 1 h before power activities can enhance performance.

**Author Contributions:** Conceptualization, D.K., Y.M. and T.M.; methodology, Y.M. and K.A.; software, D.K. and M.M.; validation, Y.M., K.A. and A.M.; formal analysis, Y.M. and D.K.; investigation, D.K.; resources, M.M.; data curation, M.M.; writing—original draft preparation, D.K.; writing—review and editing, Y.M. and T.M.; visualization, D.K. and A.M.; supervision, Y.M. and T.M. All authors have read and agreed to the published version of the manuscript.

**Funding:** This research received no external funding.

**Institutional Review Board Statement:** The study was conducted according to the guidelines of the Declaration of Helsinki, and approved by the Institutional Review Board of the School of Physical Education and Sport Science (66/2021).

**Informed Consent Statement:** Informed consent was obtained from all subjects involved in the study.

**Data Availability Statement:** The data presented in this study are available on request from the corresponding author.

**Conflicts of Interest:** The authors declare no conflict of interest.

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
