# Peer review of "Comparison of Ergogenic Effects of Caffeine and Nitrate Supplementation on Speed, Power and Repeated Sprint Performance of Soccer Players"

_physiologia, doi:10.3390/physiologia1010002_

Round 1
Reviewer 1 Report
Although the study is interesting and applicable, I have some concerns:
Please, provide a better justify for comparing the two supplements;
Also, why soccer were choose?
Experimental design should be better explained, maybe a figure would help;
Justify the doses used for caffeine and for nitrate? is there any type of equalization for these dose, since a dose-response could be different.
Provide a sample estimation calculation, since a crossover model with only 10 could hard finding differences;
Is Illinois agility test choose for any reason? Considering soccer performance, YOYO tests are more related with performance.
Since applied test are very sensitive regarding a short time and possible measurement errors, why the authors would expect differences? Could the possible differences due to test error of measurements? This point must be better clarified, because it could be assumed that any differences occurred not due to supplements use!
Why not use effect size, in addition to traditional statistics tests?
In the conclusion section the authors stated: ", the two supplements limited the decrease in performance in the test of repeated sprints with caffeine however showing a greater effect". Where is the data supporting the statement?
Author Response
Cover letter
We explain you point by point the changes on your comments
Although the study is interesting and applicable, I have some concerns:
Please, provide a better justify for comparing the two supplements;
We add this part:
… As mentioned above the performance in soccer depends on the actions taken at high intensity. That is why it is especially important to maintain the capacity for high-intensity action throughout the soccer match. Both caffeine and nitrates have been reported to improve performance in anaerobic effort…..
Also, why soccer were choose?
Our laboratory is particularly interested in improving the performance of soccer players. Also, the studies in soccer that use nitrates as supplements are limited.
Experimental design should be better explained, maybe a figure would help;
We add the next figure:
Justify the doses used for caffeine and for nitrate? is there any type of equalization for these dose, since a dose-response could be different.
However we do not know any way to compare the two doses. It is also known that different doses can have different effects. With the addition of the next sentence in the text, we report that for both supplements doses were used, in previous studies showed ergogenic action.
We add this phrase: Both the caffeine dose, and the nitrate dose used have been reported to have ergogenic effects on athletic performance.
Provide a sample estimation calculation, since a crossover model with only 10 could hard finding differences;
Αs mentioned in the last paragraph of the text the limited sample of the study is its limitation. However, it is something we cannot change now
Is Illinois agility test choose for any reason? Considering soccer performance, YOYO tests are more related with performance.
Illinois was chosen because it is a test of agility that is widely used by soccer players. The purpose of the test is to measure the ability to change direction (agility), a movement characterized by power. Yo-Yo test is also used in soccer players, however the purpose of its use is to assess the aerobic capacity of players and not movements that are affected by power.
Since applied test are very sensitive regarding a short time and possible measurement errors, why the authors would expect differences? Could the possible differences due to test error of measurements? This point must be better clarified, because it could be assumed that any differences occurred not due to supplements use!
The aim of the study was to investigate the effect of supplements on the performance of power and speedc indicators (such as speed, jump, agility). Therefore, were used these fitness tests which evaluate these indicators and are intense and have short duration. These tests have been used for years without questioning for their reliability.
Why not use effect size, in addition to traditional statistics tests?
As you can see in ‘’Statistical Analysis’’ we mentioned that: Effect sizes (ES) were estimated by calculating partial eta squared and were classified as small (0.01 to 0.058), medium (0.059 to 0.137) or large (0.138 or higher) according to Cohen (1988).14 So, we use effect size!
In the conclusion section the authors stated: ", the two supplements limited the decrease in performance in the test of repeated sprints with caffeine however showing a greater effect". Where is the data supporting the statement?
As you can see in results:
However, for the performance decrement during RSA test observed significant differences between treatments (F= 2.153, P= 0.048, η2= 0.301). More specifically the differences were observed between PG and CG (P=0.012) and between PG and NG (P=0.035).

Reviewer 2 Report
The manuscript is very interesting
Author Response
Thank you for your comments!
Round 2
Reviewer 1 Report
all comments were addressed